# Robotic vs. Transsternal Thymectomy: A Single Center Experience over 10 Years

**DOI:** 10.3390/jcm10214991

**Published:** 2021-10-27

**Authors:** Luis Filipe Azenha, Robin Deckarm, Fabrizio Minervini, Patrick Dorn, Jon Lutz, Gregor Jan Kocher

**Affiliations:** 1Division of General Thoracic Surgery, Bern University Hospital, University of Bern, 3010 Bern, Switzerland; luis.azenha@extern.insel.ch (L.F.A.); robin.deckarm@unibe.ch (R.D.); patrick.dorn@insel.ch (P.D.); jon.lutz@insel.ch (J.L.); 2Department of Thoracic Surgery, Cantonal Hospital Lucerne, 6000 Lucerne, Switzerland; Fabrizio.minervini@luks.ch

**Keywords:** RATS, robotic, thymectomy, anterior mediastinal tumor resection

## Abstract

Introduction: Thymomas are the most common tumors of the mediastinum. Traditionally, thymectomies have been performed through a transsternal (TS) approach. With the development of robot-assisted thoracic surgery (RATS), a promising, minimally invasive, alternative surgical technique for performing a thymectomy has been developed. In the current paper, the oncological and surgical outcomes of the TS vs. RATS thymectomies are discussed. Methods: For the RATS thymectomy, two 8 mm working ports and one 12 mm camera port were used. In the transsternal approach, we performed a median sternotomy and resected the thymic tissue completely, in some cases en bloc with part of the lung and/or, more frequently, a partial pericardiectomy with consequent reconstruction using a bovine pericardial patch. The decisions for using the TS vs. RATS methods were mainly based on the suspected tumor invasion of the surrounding structures on the preoperative CT scan and tumor size. Results: Between January 2010 and November 2020, 149 patients were submitted for an anterior mediastinal tumor resection at our institution. A total of 104 patients met the inclusion criteria. One procedure was performed through a hemi-clamshell incision. A total of 81 (78%) patients underwent RATS procedures, and 22 (21.1%) patients were treated using a transsternal (TS) tumor resection. Thymoma was diagnosed in 53 (51%) cases. In the RATS group, the median LOS was 3.2 ± 2.8 days and the median tumor size was 4.4 ± 2.37 cm compared to the TS group, which had a median LOS of 9 ± 7.3 days and a median tumor size of 10.4 ± 5.3 cm. Both differences were statistically significant (*p* < 0.001). Complete resection was achieved in all patients. Conclusion: While larger and infiltrating tumors (i.e., thymic carcinomas) were usually resected via a sternotomy, the RATS procedure is a good alternative for the resection of thymomas of up to 9.5 cm, and the thymectomy is a strong approach for myasthenia gravis. The oncological outcomes and survival rates were not influenced by the chosen approach.

## 1. Introduction

Mediastinal tumors are most often asymptomatic and first diagnosed using a routine chest radiograph or computed tomography. Primary tumors of the anterior mediastinum account for half of the mediastinal tumors, and the most common neoplastic lesions in the anterior mediastinum are thymoma, teratoma, and lymphoma. Other benign conditions are frequent, such as thymic hyperplasia or thymic cysts [1]. Thymic tumors are rare, malignant tumors that represent 0.2–1.5% of all malignancies. Nevertheless, they are the most common mediastinal tumors, accounting for 20% of all mediastinal tumors and 50% of anterior mediastinal tumors [2]. About 30% of thymomas are associated with Myasthenia gravis (MG), but only 10–12% of MG cases are associated with thymoma [3,4,5].

The standard surgical approach for the resection of a mediastinal tumor has been a sternotomy for many years because it offers a good overview of the anterior mediastinum and is considered safe in terms of intraoperative complications [5,6]. This comes at a heavy price for the patient when considering morbidity, such as postoperative pain, blood loss, and surgical site infection, as well as the length of stay [6,7,8]. The decrease in morbidity—thanks to the development of minimally invasive techniques as early as 1992 with VATS, followed by the introduction of the RATS thymectomy—has not come at the cost of oncological or survival outcomes [9,10,11,12]. In order to overcome the technical challenges of the minimally invasive resection of mediastinal tumors, the use of RATS seems like a logical choice. The main tumor characteristics that have been determining factors in the choice of surgical approach were tumor size and the infiltration of surrounding structures. According to some studies, a tumor size over 5 cm is not suitable for a minimally invasive approach [10,13,14]. Other authors have reported that tumor sizes of up to 9.5 cm—with and without the invasion of adjacent structures, such as the pericardium, the lung, and the phrenic nerve—were suitable for RATS resection [7].

In the current paper, we report our 10-year experience with the surgical treatment of mediastinal masses, including thymomas, thymic carcinomas, and benign thymic neoplasms.

## 2. Materials and Methods

We retrospectively reviewed the charts of 149 consecutive patients that were submitted for the resection of an anterior mediastinal mass between January 2010 and November 2020. Transsternal (TS), as well as robot-assisted (RATS) approaches, were included in the study. Surgery was performed at the Division of General Thoracic Surgery at the University Hospital of Bern, Inselspital, in Bern Switzerland.

Inclusion criteria were the resection of an anterior mediastinal tumor, primary surgery at our institution during the time period of January 2010 and December 2021, no loss to follow-up, and complete clinical and histopathological data.

Exclusion criteria were recurrence at the time of surgery, primary surgery at another institution, the use of a surgical approach other than TS/hemi-clamshell or RATS, loss to follow-up, and the presence of missing data.

All patients signed an informed consent form for study participation, and the study was approved by our internal review board (approval number, TS03-2021; date of approval, 10 May 2021).

Information on patient characteristics, postoperative histology, tumor size, tumor stage, surgical approach, intraoperative blood loss, operative time, length of stay, postoperative complications, and follow-up data, such as recurrence, was collected.

Surgical technique:

Robotic surgery:

We used a 3-port approach and the DaVinci Si robotic system (Intuitive Surgical, Sunnyvale, CA). First, a 12 mm port was inserted at the anterior axillary line in the fifth intercostal space. After an inspection of the anatomy of the chest and mediastinum, two additional 8 mm working ports were inserted in the fifth and third intercostal spaces, at the midclavicular and anterior axillary lines, respectively. Our preferred approach was from the left side, but for cases in which the main mediastinal mass was situated on the right to the midline, a right-sided approach was used. Before starting the dissection, CO_2_-Insufflation was initiated with a pressure of 8 to 10 mmHg. On the right, bipolar forceps were used (Maryland), and Cadière forceps were introduced on the left. The general surgical approach consisted of a complete thymectomy, including both upper horns, both lobes, and all of the mediastinal fat between both phrenic nerves, following a non-touch technique concerning the tumor (to avoid the rupture of the tumor capsule). The specimen was retrieved using an endobag, usually through an incision in the third intercostal space. For bigger tumors, the incision was enlarged. A 20 Fr chest tube was inserted through the lateral incision at the fifth intercostal space, and a suction pressure of 20 cmH_2_O was applied. The chest tube was removed on postoperative day 1 if no air leak was documented and fluid drainage was inferior to 200 mL.

Transsternal approach:

Surgical access was created using either a median sternotomy or a hemi-clamshell incision in the case of particularly large and infiltrating tumors. The tumor tissue and mediastinal fat were resected in toto following a no-touch policy, as discussed above. In the case of suspected infiltration, en-bloc extra-anatomical lung resection and pericardial resection were performed. In cases of pericardial resection, the pericardium was reconstructed using a bovine or porcine pericardial patch.

Classification:

The Masaoka–Koga staging system was used to define the clinical stage, and the WHO classification served as a tool to define the histological type.

Statistics:

Categorical variables were compared using chi-squared and Fischer’s exact tests. Continuous variables were compared using Mann–Whitney U tests. Data are reported as means and SDs, and statistical calculations were made using SPSS Version 25.0 (www.ibm.com, last accessed 5 July 2021) IBM^®^.

## 3. Results

Between January 2010 and December 2020, we performed 149 resections for anterior mediastinal tumors. A total of 45 patients were either lost to follow-up or did not meet the inclusion criteria. Of these 104 resections, 77.9% (*n* = 81) were performed through minimally invasive surgery using the DaVinci Si robotic platform (Intuitive Surgical, Sunnyvale, CA, USA). Tumor resection via a median sternotomy was performed in 22 patients (21.1%), mainly for thymic carcinomas (*n* = 3; 2.86%) or large thymomas (median: 10.5 ± 5.3 cm; range 4.8–23 cm).

Patient characteristics were comparable in both groups and are summarized in Table 1.

A total of 26 patients presented with MG symptoms at the initial evaluation. After the thymectomy, only five patients remained symptomatic and required medical treatment for myasthenia gravis. Three patients with persisting MG symptoms were treated using a RATS approach, and two were treated using a TS approach. Histopathological examinations revealed thymoma in 18 cases and thymic hyperplasia in 8 cases; this corresponds to 33% of patients diagnosed with thymoma. All patients with persisting symptoms had histopathologically confirmed thymoma.

Three patients in the TS group received neoadjuvant chemotherapy. Fine-needle aspiration biopsies revealed one case of thymoma and two cases of thymic carcinoma. None of the patients in the RATS group were treated with neoadjuvant chemotherapy.

A hemi-clamshell incision was performed in one patient (1%) with a large thymoma invading the right hemithorax. The median blood loss during the RATS thymectomy was 45 mL ± 24.71 (range 20–600 mL), as opposed to 435 mL ± 313.1 during the TS thymectomy (range 100–1400 mL) *p* < 0.001. The median operative time for the TS thymectomy was 173.6 ± 86 min and 95 ± 29 min for the RATS thymectomy *p* < 0.001. Operative data are presented in Table 2.

Histology showed benign lesions in 41.3% (*n* = 43), thymoma in 51% (*n* = 53), and thymic carcinoma in 3.8% (*n* = 4) of patients. Additionally, histology revealed teratoma in 2.9% (*n* = 3) and lymphoma in 1% (*n* = 1) of patients, as shown in Table 3. The RATS approach was used for tumors with a median size of 4.4 ± 2.4 cm (including thymomas), and the median LOS of these patients was 3.2 ± 2.8 days (range 1–12 days). On the other hand, the mean LOS for patients operated on using a transsternal approach was 9.9 ± 7.3 days (range 4–33). The difference in LOS between the TS and RATS tumorectomies is statistically significant (*p* = 0.002).

The TS tumorectomy group had a higher rate of postoperative complications requiring therapy: 4.8% (*n* = 5) for the minimally invasive vs. 9.6% (*n* = 13) for the transsternal approaches. Overall, the complication rate was too low to reach a statistically significant difference.

The main complications included bleeding (*n* = 2), pneumonia (*n* = 2), chylothorax (*n* = 1), myasthenic crisis (*n* = 1), pericardial effusion (*n* = 2), intestinal perforation (*n* = 2), surgical site infection (*n* = 2), urosepsis (*n* = 1), and phrenic nerve paralysis (*n* = 2), as shown in Table 4.

In patients with thymoma, the Masaoka tumor stages were: I in 22 patients (42%), IIa in 19 patients (35.6%), IIb in 7 patients (13.2%), III in 3 patients (5.6%), and IVa in 2 patients (3.6%). There was no statistically significant difference in tumor stage between the RATS and TS approaches (*p* = 0.77).

Complete resection was achieved in all patients across both groups, and the conversion rate (from robotic to thoracotomy or sternotomy) was 0%. The resection of the surrounding structures was performed for five patients in the RATS group and seven patients in the TS group (*p* = 0.091) (Table 5).

After a mean follow-up of 44 months (range 1–108), *n* = 4 patients with thymoma showed recurrence, 3 were treated via a transsternal approach, and only 1 was treated via a RATS approach. In the group with thymic carcinoma, 1 patient showed pleural recurrence after 14 months and was treated through a minimally invasive resection of the recurrence sites. Five patients died during follow-up; only -one of these deaths was due to tumor progression.

## 4. Discussion

This study shows that the RATS thymectomy approach for the treatment of tumors up to 9.5 cm in size is technically feasible and safe. Neither complete resection rates nor oncological outcomes were compromised by the minimally invasive nature of the approach. Complete resection has been known to be a major prognostic factor for survival in thymic surgery and is not negatively affected by the RATS approach [5,12,15,16,17]. Two recent studies conducted by Kneuertz et al. and Wilshire et al. published similar data—with reported complete and safe resections of tumors of up to 9.5 cm in size—including tumors that had infiltrated surrounding structures, such as the lung, the pericardium, and the phrenic nerve [7,14]. Our results confirm these findings and emphasize that neither tumor size nor tumor infiltration of surrounding structures are clear contraindications for a minimally invasive approach. This is in contrast to previous publications stating that RATS was only suitable for lesions of up to 5 cm [13,14].

The magnified 3D vision of the RATS method allows for the meticulous dissection of the tumor and its surrounding structures as well as the identification of nourishing vessels arising from large vascular structures, such as the brachiocephalic trunk. The use of bipolar diathermy provides additional safety and facilitates close dissection to the phrenic nerve. Tumor retrieval can be difficult but is feasible using a solid endobag and a slight enlargement of the incision, if necessary. In case of very large and solid tumors, one of the robotic arms can be introduced through a subxiphoidal incision, which can easily be enlarged enough to allow for the retrieval of such specimens. This technique was used in one patient with a tumor of 9.5 cm. A left-sided approach provides, in our opinion, a better overview of the mediastinum and the most delicate structures, such as the brachiocephalic vein and both phrenic nerves. Due to the left-sided shift of the heart, vision is less obstructed and a better overview is guaranteed.

Our cohort revealed that 18 patients out of 53 diagnosed with thymoma suffered from myasthenia gravis symptoms. This corresponds to 33% percent of patients with histopathologically confirmed thymoma and is in accordance with previously published studies [3,4,5]. This could be explained by the presence of ectopic thymic tissue. At our clinic, we routinely screen patients with suspicion of thymic neoplasia for the presence of acetylcholine receptor (AChR) antibodies. In the case of the clinical manifestation of MG or the presence of AChR antibodies, the patient is referred to a neurologist for evaluation and the establishment of medical treatment, if necessary. After at least four weeks of medical treatment and clinically controlled MG symptoms, we proceed to the procedure. Perioperatively, steroids are administered. One patient developed a myasthenic crisis even though he received a neurological workup and treatment before surgery.

As shown by previous studies, we can confirm that the RATS thymectomy is associated with less perioperative and postoperative morbidity in terms of blood loss, pneumonia, postoperative bleeding, and surgical site infection. The most frequent postoperative complication occurring after a RATS thymectomy was postoperative pericardial effusion and urinary retention. The latter is a common complication after general anesthesia, especially in men. Pericardial effusion occurred more often in the RATS groups, which can be explained by the fact that the pericardium was less frequently perforated or resected during the RATS approach. This directly translates into shorter lengths of stays and earlier chest tube removal [6,8,10,18,19,20]. Masaoka tumor stages and histological type were similar among both groups, without any difference in outcomes. This supports the affirmation of previous studies reporting no differences in oncological outcomes between the TS and minimally invasive thymomectomies, especially in early-stage thymomas [9,10,20].

Only three patients in the TS group, and none in the RATS group, received neoadjuvant chemotherapy. This low rate can be explained by our department’s policy to resect mediastinal tumors without previous biopsy if complete (R0) resection seems feasible, in order to avoid tumor spilling and spreading from a transthoracic biopsy.

In our series, 50% of resected tumors were identified as thymomas, which is in accordance with previous publications and suggests an accurate case selection [1]. Although en bloc resection of the mediastinal tumor is feasible using minimally invasive surgery, it has its limits. TS tumor resection still has its place in the resection of very large tumors or tumors infiltrating the surrounding structures to a large extent. In our study population, large, extended resections of adjacent structures, such as the brachiocephalic vein and lung parenchyma, were performed solely via a transsternal approach. A TS approach also facilitates the reconstruction of resected structures, such as the pericardium and large vessels (e.g., the brachiocephalic vein or superior vena cava) in case of tumor infiltration [20]. The careful study of preoperative imaging is mandatory in order to maintain a low intraoperative conversion rate and choose the most adequate procedure for every single patient.

Concerning the learning curve for robotic surgery, it can be stated that with proper step-wise training, the technique is easily applicable.

## 5. Conclusions

In conclusion, our study demonstrates that a robotic resection is completely safe and feasible, even for large mediastinal tumors up to 9.5 cm in size, without compromising oncological outcomes. Nevertheless, the transsternal approach still has its role in the treatment of locally advanced and very large tumors.

## Figures and Tables

**Table 1 jcm-10-04991-t001:** Patient characteristics.

	RATS	TS	Significance
Age (years)	Median: 51 Standard Deviation: 19.3 Range: 18–84	Median: 54 Standard Deviation: 17.1 Range: 21–80	*p* = 0.426
Gender	Male: 42 Female: 39	Male: 12 Female: 11	*p* = 0.279

**Table 2 jcm-10-04991-t002:** Operative data.

	RATS	TS	Significance
Operative Time (min)	Median: 95 Standard Deviation: 29.1 Range: 48–204	Median: 173 Standard Deviation: 86 Range: 84–430	*p* < 0.001
Blood Loss (ml)	Median: 45 Standard Deviation: 25 Range: 20–600	Median: 435 Standard Deviation: 313 Range: 100–1400	*p* < 0.001
Tumor Size (cm)	Median: 4.4 Standard Deviation: 2.37 Range: 1.5–9.5	Median: 10.4 Standard Deviation: 5.3 Range: 4.8–23	*p* = 0.011

**Table 3 jcm-10-04991-t003:** Histopathological results.

Histology	RATS (*n*)	TS (*n*)
Thymoma	34	19
Thymic Carcinoma	1	3
Teratoma	2	1
Thymic Hyperplasia	32	0
Thymic Cyst	11	0
Lymphoma	1	0

**Table 4 jcm-10-04991-t004:** Postoperative complications.

	TS (*n*)	RATS (*n*)
Bleeding	2	0
Intestinal Perforation	2	0
Phrenic Nerve Paralysis (Left)	2	0
Pneumonia	2	0
Respiratory Insufficiency	1	0
Surgical Site Infection	2	0
Urosepsis	1	0
Chylothorax	0	1
Atrial Fibrillation	1	0
Urinary Retention	0	2
Pericardial Effusion	0	2

**Table 5 jcm-10-04991-t005:** Resected adjacent structures.

	TS(*n*)	RATS (*n*)
Pericardium	7	5
Lung	3	0
Phrenic Nerve	1	0
Breachiocephalic Vein	3	0

## Data Availability

Detailed Data can be obtained from the corresponding author upon request.

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
