# Peer review of "Robotic vs. Transsternal Thymectomy: A Single Center Experience over 10 Years"

_jcm, 2021, doi:10.3390/jcm10214991_

Round 1

Reviewer 1 Report

The authors reported their experience in mediastinal surgery by robotic approach, evaluating the post-operative results and comparing them with the outcomes of the trans- sternal approach.

The paper is  interesting and well written , showing a well-conducted study.

I have some comments

  • In line 50, the tumour size is indicated as the main characteristic for evaluating the better surgical approach to choice. According to NCCN guidelines, minimally invasive surgery may be considered for clinical stage I-II. Furthermore, previous studies demonstrated the feasibility and safety of robotic thymectomy in case of large lesion(1,2). These information should be reported and commented on.
  1. Wilshire CL, Vallières E, Shultz D, Aye RW, Farivar AS, Louie BE. Robotic Resection of 3 cm and Larger Thymomas Is Associated With Low Perioperative Morbidity and Mortality. Innovations (Phila). 2016 Sep/Oct;11(5):321-326.
  2. Kneuertz PJ, Kamel MK, Stiles BM, Lee BE, Rahouma M, Nasar A, Altorki NK, Port JL. Robotic Thymectomy Is Feasible for Large Thymomas: A Propensity-Matched Comparison. Ann Thorac Surg. 2017 Nov;104(5):1673-1678
  • The myasthenic crisis was described among the complications, but the rate of patients with thymic lesion and myasthenia gravis is not reported by Authors . This information and also the evaluation of the outcomes of myasthenic patients, according to the different surgical approach could be inserted
  • What was the rate of patients with thymic neoplasm who underwent neo-adjuvant therapy? Were steroids administered to thymoma patients in the pre-operative period?
  • Post-operative complications were briefly described. A table showing the specific complications and the statistical difference in the two groups (robotic vs open surgery) could be interesting .

Author Response

Dear reviewer,

thank you for your kind comments.  We revised the article according to the issues you pointed out. Hereunder you will find a more detailed description point by point, as well as the line in the text where you can find the modifications.

  • In line 50, the tumour size is indicated as the main characteristic for evaluating the better surgical approach to choice. According to NCCN guidelines, minimally invasive surgery may be considered for clinical stage I-II. Furthermore, previous studies demonstrated the feasibility and safety of robotic thymectomy in case of large lesion(1,2). These information should be reported and commented on.
  1. Wilshire CL, Vallières E, Shultz D, Aye RW, Farivar AS, Louie BE. Robotic Resection of 3 cm and Larger Thymomas Is Associated With Low Perioperative Morbidity and Mortality. Innovations (Phila). 2016 Sep/Oct;11(5):321-326.
  2. Kneuertz PJ, Kamel MK, Stiles BM, Lee BE, Rahouma M, Nasar A, Altorki NK, Port JL. Robotic Thymectomy Is Feasible for Large Thymomas: A Propensity-Matched Comparison. Ann Thorac Surg. 2017 Nov;104(5):1673-1678

Answer: We reported the findings of these studies and compared them to ours (line 203 to 207)

  • The myasthenic crisis was described among the complications, but the rate of patients with thymic lesion and myasthenia gravis is not reported by Authors . This information and also the evaluation of the outcomes of myasthenic patients, according to the different surgical approach could be inserted

Answer: This question was addressed, and our SOP described in the results and discussion section (lines 139 to 144; 270 to 276). The presence of MG had no effect on the choice of technique in this cohort.

  • What was the rate of patients with thymic neoplasm who underwent neo-adjuvant therapy? Were steroids administered to thymoma patients in the pre-operative period?

Answer: we reviewed the data and came to the conclusion that only 3 patients received neoadjuvant chemotherapy (line146-148) . We commented the low rate of neoadjuvant chemotherapy in the discussion section (line 280 to 283).

Steroids were administered after neurological assessment in patients presenting MG symptoms. We added this information in line 225 to 235

  • Post-operative complications were briefly described. A table showing the specific complications and the statistical difference in the two groups (robotic vs open surgery) could be interesting .

Answer: A table was added on page 5 line 177. The complication rate is too low to perform a statistical analysis.

We hope these explanation will meet your exception.

Kind regards

Reviewer 2 Report

To the authors ,

the article is well written and interesting. We appreciate very much that you have divided your cohort of patients : patients with a benign pathology and those suffering for thymoma. You have mentioned the rate of people with thymus pathology affected by Miastenia Gravis: how many patients on your study were affected by MG ? Which kind of surgical technique have you chosen for this kind of patient and why? We agree with you on the choice of the left side. I think that you can describe the anatomical reason ,or the surgical technique motivation for this choice because it could be a discussed topic.  Have you talk about the postoperative complications : can you describe which were the most commons one for the robotic technique and why ?  

Author Response

Dear reviewer, 

Thank you for your kind and constructive comments. Herunder you will find a detailed reply to your requests as well as the line you can find them on the main document. 

You have mentioned the rate of people with thymus pathology affected by Miastenia Gravis: how many patients on your study were affected by MG ? Which kind of surgical technique have you chosen for this kind of patient and why?

Answer: This question was addressed, and our SOP described in the results and discussion section (lines 139 to 144; 270 to 276). The presence of MG had no effect on the choice of technique in this cohort.

We agree with you on the choice of the left side. I think that you can describe the anatomical reason ,or the surgical technique motivation for this choice because it could be a discussed topic. 

Answer: This topic was addressed in the discussion line 220 to 223

You talked about the postoperative complications : can you describe which were the most common ones for the robotic technique and why ? 

Answer: A table was added on page 5 line 177 Overall the complication rate is too low to perform a statistical analysis. The explanation for the most common complication was added to the discussion section line 228 to 233

We hope the modifications meet your expectations.

Kind regards

Reviewer 3 Report

This paper presented the results of transsternal versus robot assisted thymectomy at a single-institution. It compared the two outcomes and did find significant difference. This is a very interesting topic and robot-assisted thymectomies are becoming more commonplace. However, this is a poor comparison. The two cohorts have clear selection bias, being that the trans-sternal thymectomy tumors tended to be larger and more invasive, it is likely to have more blood loss, longer operative time etc. I think that if you are to present your results from robotic thymectomies, it is important to know just those specific outcomes, the demographics of those specific patients. Should be compared to patients of similar status. 

Author Response

Dear reviewer,

Thanks for your feedback. The aim of this study was not to statistically prove that robotic is better than transsternal surgery. We rather present a descriptive study, pointing out the differences between the two patient cohorts. Our goal was to show that robotic surgery is feasible and safe for tumors up to 9.5cm in size and has comparable oncological outcomes. To answer your request a randomized controlled trial would be necessary. Considering today’s standards this would be ethically questionable, so we must rely on the data we have.

Kind regards.

The authors